# Examining the Factor Structure of the Pittsburgh Sleep Quality Index in a Multi-Ethnic Working Population in Singapore

**DOI:** 10.3390/ijerph16234590

**Published:** 2019-11-20

**Authors:** Gerard Dunleavy, Ram Bajpai, André Comiran Tonon, Ai Ping Chua, Kei Long Cheung, Chee-Kiong Soh, Georgios Christopoulos, Hein de Vries, Josip Car

**Affiliations:** 1Centre for Population Health Sciences, Lee Kong Chian School of Medicine, Nanyang Technological University Singapore, 11 Mandalay Road, Singapore 308232, Singapore; rambajpai@hotmail.com (R.B.); josip.car@ntu.edu.sg (J.C.); 2Department of Health Promotion, CAPHRI Care and Public Health Research Institute, Maastricht University, P.O. Box 616, 6022 MD Maastricht, The Netherlands; hein.devries@maastrichtuniversity.nl; 3Research Institute for Primary Care and Health Sciences, Keele University, David Weatherall Building, Staffordshire ST5 5BG, UK; 4Laboratório de Cronobiologia e Sono, Porto Alegre Clínicas Hospital (HCPA), R. Ramiro Barcelos, 2350—Santa Cecilia, Porto Alegre 90035-007, RS, Brazil; andrectonon@gmail.com; 5Postgraduate Program in Psychiatry and Behavioral Sciences, Federal University of Rio Grande Do Sul (UFRGS), Av. Paulo Gama, 110—Farroupilha, Porto Alegre 90040-060, RS, Brazil; 6Department of Medicine, Jurong Health Campus, National University Health System, 1 Jurong East Street 21, Singapore 609606, Singapore; ai_ping_chua@nuhs.edu.sg; 7Department of Clinical Sciences, College of Health and Life Sciences, Brunel University London, Kingston Lane, Uxbridge, Middlesex, London UB8 3PH, UK; KeiLong.Cheung@brunel.ac.uk; 8School of Civil and Environmental Engineering, College of Engineering, Nanyang Technological University Singapore, 50 Nanyang Avenue, Singapore 639798, Singapore; CSOHCK@ntu.edu.sg; 9Division of Leadership, Management and Organisation, Nanyang Business School, College of Business, Nanyang Technological University Singapore, 50 Nanyang Avenue, Singapore 639798, Singapore; CGeorgios@ntu.edu.sg; 10Department of Primary Care and Public Health, School of Public Health, Imperial College London, London SW7 2AZ, UK

**Keywords:** sleep quality, factor analysis, workplace health

## Abstract

The Pittsburgh Sleep Quality Index (PSQI) is a widely used measure for assessing sleep impairment. Although it was developed as a unidimensional instrument, there is much debate that it contains multidimensional latent constructs. This study aims to investigate the dimensionality of the underlying factor structure of the PSQI in a multi-ethnic working population in Singapore. The PSQI was administered on three occasions (baseline, 3 months and 12 months) to full-time employees participating in a workplace cohort study. Exploratory factor analysis (EFA) investigated the latent factor structure of the scale at each timepoint. Confirmatory factor analysis (CFA) evaluated the model identified by EFA, and additionally evaluated it against a single factor and a three-factor model. The EFA identified a two-factor model with similar internal consistency and goodness-of-fit across each timepoint. In the CFA, the two- and three-factor models were both superior to the unidimensional model. The two- and three-factor models of the PSQI were reliable, consistent and provided similar goodness-of-fit over time, and both models were superior to the unidimensional measure. We recommend using the two-factor model to assess sleep characteristics in working populations in Singapore, given that it performs as well as the three-factor model and is simpler compared to the latter.

## 1. Introduction

Sleep, essential for the physical repair and restoration of the body [1], is influenced by several internal and external factors. The transformation of cities towards 24-hour societies provides a major challenge to people attaining the sleep duration recommended for optimal health. Ever-increasing urbanization has led to increases in the density and distribution of artificial lighting, with more than 80% of the world’s population living under light-polluted skies. Light pollution is associated with both sleep deprivation and sleep disorders [2,3]. This should be of concern to residents of Singapore, as not only is the city-state the most light-polluted country in the world [4], it is also reported to be home to the world’s most sleep-deprived population [5]. A growing body of evidence has highlighted the relationship between poor sleep and a variety of chronic conditions such as diabetes, hypertension, heart disease, stroke, and cancer, as well as mood disorders including anxiety and depression [6]; it is, therefore, paramount that sleep quality is accurately measured and monitored.

The Pittsburgh Sleep Quality Index (PSQI) is a well-validated sleep quality questionnaire [7]. The PSQI is the most commonly used sleep assessment instrument across clinical and research settings [8]. The tool encompasses seven components: subjective sleep quality, sleep latency, sleep duration, habitual sleep efficiency, sleep disturbances, use of sleeping medications, and daytime dysfunction, which are summed together to make up an overall sleep quality score. Individuals are then defined as “good” or “poor” sleepers based on their sleep quality score being ≤5 or >5, respectively. Given the multifaceted nature of sleep and the unique aspects of sleep represented by the seven individual components of the PSQI, the dichotomization of the sleep quality score means that individuals who perform negatively on one or two components but score better on the remaining components may be defined as a “good” sleeper, and go undetected for the sleep issues that they are experiencing. Hence, it is much debated whether the unidimensional application of the instrument is adept to comprehensively evaluate the multifaceted nature of sleep.

While some studies support the unidimensional application of the instrument, a recent systematic review found that the majority of studies conclude that the instrument is best viewed as a multidimensional tool [9]. The factor structure of the PSQI may differ depending on the population in which it is administered, hence the need to examine its dimensionality across populations, to determine its suitability and best application. The vast majority of studies examining the factor structure of the PSQI have been performed in specific clinical populations, most commonly in western countries [10,11,12,13,14,15], with few conducted with samples in South-East Asia [16]. A study examining the factor structure of the PSQI among the general population in Singapore proposed that a three-factor model (“sleep efficiency”, “perceived sleep quality”, and “daily disturbances”) was superior to the unidimensional model originally proposed by Buysse et al. [7]. However, the evidence so far on the distinct presentation of impaired sleep quality in a working population in Singapore is lacking, and given that sleep quality may vary between working and general populations [17], further research is needed. Therefore, the aim of this study is to explore the factor structure of PSQI prospectively and its consistency over time among a multi-ethnic working population in Singapore, as well as its comparison with previously suggested models. 

## 2. Materials and Methods

### 2.1. Study Design and Sample

A prospective cohort study (2016–2018) was conducted at four workplaces in Singapore. Details of the cohort study design are published elsewhere [18]. Briefly, volunteers from four workplaces, consisting of two from the transport industry, a cooling plant, and a university, were invited to participate in the study via meetings, workplace posters, and emails. A total of 464 full-time employees (age ≥21 years) were recruited and enrolled into the study. After 3 months from baseline, 424 (91.4%) participants completed follow-up, and after 12 months from baseline, 334 (72.0%) completed follow-up. The PSQI was administered to the participants at the point of enrolment (timepoint 1), and at 3 and 12 months following enrolment (timepoints 2 and 3, respectively). This sample size range was adequate, as it exceeds the minimum of 250 recommended for covariance structure modeling analysis [19,20].

### 2.2. Measurements

Sleep quality was measured using the PSQI [7]. This questionnaire has 19 self-rated items grouped into seven components: subjective sleep quality, sleep latency, sleep duration, habitual sleep efficiency, sleep disturbances, sleep medication use, and daytime dysfunction. Each of these seven components are equally weighted based on a 0 to 3 scale, whereby 3 reflects the negative extreme on the Likert Scale. The subscales are aggregated into a global PSQI score ranging from 0 to 21, with higher scores indicating poorer sleep quality and a global sum over 5 indicates a “poor” sleeper [7].

Data on age, sex (male and female), marital status (never married, divorced, widowed and married), education (primary and secondary, pre-college, and college degree and above), occupation, nationality (Singaporean or foreigner), ethnicity (Chinese, Malay, Indian, or others), housing type (HDB flat, condominium, terrace, semi-detached, or bungalow) and monthly income (<S$2000, S$2000–S$3999, S$4000–S$5999, S$6000–S$9999, ≥S$10,000) were also collected.

### 2.3. Statistical Analysis 

We first explored frequency distributions of demographic characteristics of participants at each timepoint. Participants’ characteristics were summarized using mean ± standard deviation or median (interquartile range [IQR]) for numeric variables, and categorical variables using frequency and percentage. We assessed the internal consistency of the PSQI using Cronbach’s alpha and inter-item correlations using Spearman’s rank correlation coefficients (<0.5: low; 0.5-0.7: moderate; and >0.7: high). We assessed the test-retest reliability of the global PSQI score and factors from the EFA by intra-cluster coefficient (ICC).

We assessed the suitability of the data for performing factor analysis with Bartlett’s test of sphericity (testing the null hypothesis that the correlation matrix is an identity matrix) and Kaiser-Meyer-Olkin (KMO) measure of sampling adequacy (taking test values of >0.5 as acceptable) [21]. We then explored the factor structure of the PSQI using both exploratory factor analysis and confirmatory factor analysis approaches at each of the three timepoints separately. We conducted the exploratory factor analysis (EFA) using maximum likelihood estimation with orthogonal rotation. We used the scree plot and eigenvalues (≥1) to determine the required number of meaningful factors. A rotated factor loading threshold of 0.4 was considered meaningful in identifying items with common characteristics [22,23]. 

Next, a Confirmatory Factor Analysis (CFA) using maximum likelihood estimation approaches was conducted to evaluate the model fit. To assess model fit, we calculated the delta chi-squared (Δχ^2^) statistics [24], the root mean square error of approximation (RMSEA) along with 95% confidence interval (95% CI) [25], comparative fit index (CFI), standardized root mean square residual (SRMR) goodness-of-fit index (GFI), adjusted goodness-of-fit index (AGFI), Tucker Lewis index (TLI), consistent Akaike information criterion (CAIC) and Bayesian information criteria (BIC) [26]. The RMSEA is a measure of fit that considers how much error there is for each degree of freedom. The CFI is a widely used measure that compares the model with a baseline model that assumes there is no relationship among the observed indicated variables. Despite the absence of consensus concerning the cut-off for goodness-of-fit, we elected to use the criteria recommended by Brown [25] and Kline [27]. The following criteria were suggested for reasonably good fit: 1) SRMR values ≤0.08; 2) RMSEA values ≤0.06; 3) comparative fit index ≥0.90; 4) the model chi-square (lower value will be preferred); 5) GFI ≥ 0.95; 6) AGFI ≥0.90. Furthermore, Raftery demonstrated in his paper that a difference of at least 10 in the BIC of two models would render the model with higher BIC to be rejected [26]. We also compared the models generated from the EFA against the three-factor model proposed by Koh et al. [28] and the single-factor model proposed by Buysse et al. [7] at each timepoint.

Factor loadings (i.e., the correlation between each PSQI component to each factor) were evaluated against criteria from Comrey and Lee [29]: 0.71 or greater signifies excellent loadings, 0.63 to 0.70 are very good; 0.55 to 0.62 are good; 0.45 to 0.54 are fair; and 0.32 to 0.44 are deemed poor; while any values lower than 0.32 are discarded.

Data were analyzed with Stata software (version 15.0; Stata Corp LP, College Station, TX, USA). A two-sided *p*-value <0.05 was considered for statistical significance.

### 2.4. Ethics Approval 

The study was conducted in accordance with the Declaration of Helsinki and approved by the Institutional Review Board of Nanyang Technological University Singapore, Singapore (IRB-2015-11-028). Written informed consent was obtained from all study participants prior to the commencement of data collection.

## 3. Results

### 3.1. Characteristics of Study Participants

Table 1 shows the characteristics of the population at each timepoint. At baseline, the mean age of participants was 39.0 (±11.4) years with a large proportion (41%) aged more than 40 years; the majority were male (79.5%), married (60.3%), educated up to pre-college or above (75%), and were earning <S$4000 per month (71.3%). Reflecting national representation, a large proportion were Chinese (63.8%), followed by Malays (21.3%), Indians (10.3%), and other Asian groups (4.5%). The mean global PSQI score was 5.51 (SD = 2.78), 5.43 (SD = 2.79), and 5.08 (SD = 2.77) at timepoints 1, 2 and 3, respectively. There was no statistical difference (*p* = 0.079) observed in mean global PSQI scores over time. Using the recommended cut-off of 5 for the global PSQI score, 42.5%, 43.2% and 38.3% of the participants had poor sleep quality at timepoints 1, 2 and 3, respectively.

### 3.2. Reliability and Correlation Analysis of PSQI Subscales

Table 2 provides the Spearman’s correlations and the descriptive statistics for the seven PSQI components in both studies. Among the inter-component correlations at timepoint 1, subjective sleep quality and sleep latency were found to have the highest correlation (*r* = 0.42) and daytime dysfunction and habitual sleep efficiency had the lowest correlation (*r* = 0.04). Similarly, at timepoint 3, subjective sleep quality and sleep latency were also found to have the highest correlation (*r* = 0.45), while sleep medication use had the lowest correlations (*r* = 0.01) (*r* = −0.01) (*r* = 0.01) with subjective sleep quality, sleep duration and habitual sleep efficiency, respectively. At timepoint 2, sleep disturbances and sleep latency were found to have the highest correlation (*r* = 0.40), and sleep medication use had the lowest correlations (*r* = 0.01) (*r* = −0.01) with sleep duration and habitual sleep efficiency, respectively. The ICC values for the global PSQI score across the three timepoints was 0.62, while it was 0.64 and 0.38 for the factors perceived sleep quality and sleep efficiency, respectively.

The Cronbach’s alpha of the PSQI was below desired levels at each timepoint, indicating that the scale has low reliability (T1: α = 0.63, T2: α = 0.62, T3: α = 0.62). The Cronbach’s alpha of the PSQI marginally increased when the subscale ‘sleep medication use’ was removed (T1: α = 0.65, T2: α = 0.64, T3: α = 0.66). Sleep medication use had the lowest correlation coefficient, between 0.34 and 0.41 across each time point. The low correlation of this subscale with the overall scale may reflect the low use of sleep medication in this population. None of the other subscales significantly increased overall PSQI scale (see results in Appendix A).

KMO (T1: 0.688, T2: 0.695, T3: 0.729) and Bartlett (T1: Χ^2^ = 389.373; df = 21; *p* < 0.001, T2: Χ^2^ = 330.860; df = 21; *p* < 0.001, T3: Χ^2^ = 270.947; df = 21; *p* < 0.001) tests were performed and the results verified that factor analysis was suitable [30] across each of the three timepoints.

### 3.3. Exploratory Factor Analysis

The EFA was first used to explore the underlying construct of the PSQI and the same two-factor model was produced by the EFA at each timepoint. The first factor was labeled ‘perceived sleep quality’ and the second factor was labeled ‘sleep efficiency’. Subjective sleep quality, sleep latency, sleep disturbances, sleep medication use, and daytime dysfunction loaded on the first factor, while the rest of the components had good loadings on the second factor. Table 3 shows the factor matrix for the two-factor solutions across 3 timepoints, at timepoint 1, 2 and 3, the variance accounted for by the factor perceived sleep quality was 32.1%, 31.6% and 32.8%, respectively. The variance accounted for by the second factor, sleep efficiency was 16.7%, 17.2% and 15.7%, at timepoint 1, 2 and 3, respectively. Finally, there was a small positive correlation between the two factors (*r* = 0.42) at timepoint 1, and moderate positive correlation (*r* = 0.50; *r* = 0.63) at timepoints 2 and 3, respectively (see Figure 1). The results from the sensitivity analysis involving only female participants, produced similar results, and the same two factors as the EFA for the whole sample combined. The results for the factor matrix for the two-factor solutions at each timepoint for female participants only, is provided in Appendix A.

### 3.4. Confirmatory Factor Analysis

We compared our two-factor model with the three-factor model suggested by Koh et al. and Buysse’s single-factor model. The summary of the model fits is presented in Table 4. Buysse’s single-factor model resulted in the poorest goodness-of-fit indices across each of the timepoints. The fit of the two- and three-factor models were better than the single-factor model across each timepoint. At timepoint 1, the three-factor model (χ^2^ = 28.46, *p* = 0.003; CFI = 0.95; RMSEA = 0.06; SRMR = 0.03; GFI = 0.93; AGFI = 0.86; TLI = 0.91; CAIC = 54.12; BIC = 6298.79) provided a marginally better fit than the two-factor model (χ^2^ = 36.61, p<0.001; CFI = 0.94; RMSEA = 0.06; SRMR = 0.03; GFI = 0.091; AGFI = 0.85; TLI = 0.90; CAIC = 62.28; BIC = 6294.66); however, both models provided an adequate fit to the data. Similar results were observed at timepoint 2, the three-factor model (GFI = 0.89; AGFI = 0.87; CFI = 0.92; TLI = 0.84; RMSEA = 0.08; BIC = 5957.59) was a marginally better fit than the across various fit indicies than the two-factor model (GFI = 0.87; AGFI = 0.798; CFI = 0.90; TLI = 0.84; RMSEA = 0.08; BIC = 5952.44). This trend continued at timepoint 3, the three-factor model (GFI = 0.94; AGFI = 0.88; CFI = 0.98; TLI = 0.95; RMSEA = 0.04; BIC = 4556.73) provided a similar but marginally better fit to the data than the two-factor model (GFI = 0.92; AGFI = 0.88; CFI = 0.97; TLI = 0.95; RMSEA = 0.04; BIC = 4549.17). 

## 4. Discussion

The present study examined the factor structure of the PSQI in a multi-ethnic working population in Singapore in a longitudinal fashion. The same two-factor model containing two distinct sleep quality components (labeled ‘perceived sleep quality’ and ‘sleep efficiency’) were identified by the EFA across each of the three timepoints. Consistent with an increasing body of literature, the subsequent CFA demonstrated that the PSQI should be considered as a multidimensional tool in the assessment of sleep, given that the two-factor model, identified in the EFA, and the three-factor model proposed by Koh et al. each demonstrated a superior fit to the data compared to the original single-factor model. As there was little difference between the model fit statistics of the two- and three-factor models, and considering the principle of model parsimony, we propose that the two-factor model is more appropriate to use in the evaluation of sleep quality in a working population in Singapore. The two-factor model reported from our EFA is composed of a factor labeled as “perceived sleep quality”, which consisted of the subjective sleep quality, sleep latency, sleep disturbances, sleep medication use, and daytime dysfunction subscales, and a second factor labeled as “sleep efficiency” which encompassed the remaining two subscales of the PSQI. The average global PSQI score in this study ranged between 5.1 and 5.5 over the three timepoints. This is in line with previous values observed in general population surveys in Singapore [28], and significantly lower than what is observed from clinical populations and studies with older adults in Singapore [31,32]. 

The construct of sleep quality measured by the PSQI encompasses seven components that form distinct clusters of individuals according to their health/illness status. For example, medical outpatients not only have a higher global score compared to non-clinical samples; they also vary their reports on component scores according to the types of clinics they attend and the nature of the sleep disturbances they present with [33], which can be dependent on varying causes such as medical/physical conditions, sleep disorders, mental/psychological factors, behavioral issues, environmental factors or a combination of any of the above, all of which can impact sleep adversely in different ways, and consequently impact the magnitude of sleep quality through different components. Individuals with obstructive sleep apnea (OSA), one of the most common sleep disorders, are more likely to report shorter sleep onset, longer sleep duration but yet, poorer perceived sleep quality and higher sleep disturbances and daytime dysfunction, compared to healthy controls [34]. On the other hand, individuals with primary insomnia or insomnia comorbid with mood and psychological disorders, another major sleep disorder worldwide, typically report longer sleep onset, shorter sleep duration, lower habitual sleep efficiency and more prevalent drug use to aid their sleep compared to OSA patients and healthy individuals [35]. These results reinforce the need for a multidimensional approach to the PSQI report, especially for clinical samples.

The internal consistency of the PSQI, as measured by Cronbach’s alpha, was 62% at each timepoint, indicating relatively low reliability. However, these findings are in agreement with a large population-based study from Singapore and a study from Belgium, which both reported Cronbach’s alpha of 64% [10,28]. The low internal consistency found in our sample and reported in other studies is an additional limitation of the use of the PSQI as a unidimensional measure, and further supports the use of multi-dimensional models. The original validation study of the PSQI reported a high degree of internal consistency with a Cronbach’s alpha of 83%. However, our study sample is highly heterogeneous to those enrolled in the original study, with two-thirds of participants enrolled in the original study identified as clinical cases of either mood or sleep disorders.

There was a low correlation between the subscale “sleep medication use” and the overall scale, which may reflect the low use of sleep medication in this population, with 95.7%, 94.1%, 93.9% of participants reporting that they had not used sleep medication in the previous month at timepoints 1, 2, and 3, respectively. Similarly, low correlations have been observed between this subscale and the global PSQI scale in studies of the general population [28,36,37,38]. These results highlight that while this PSQI component may be important in some clinical samples, it may not be as pertinent in epidemiological studies with otherwise healthy populations. This also indicates that model fit statistics should be tested in clinical samples with a higher prevalence of medication use.

The test-retest reliability of the PSQI instrument and the factors “perceived sleep quality” and “sleep efficiency” was lower than the desired level (≥.80) for widely used instruments [39]. Given that adequate test-retest reliability of the PSQI has been reported in previous studies with shorter time-gaps between assessments (2–45 days) [14,40,41], and the potential for sleep quality to change, it is possible that the low test-retest reliability observed in this study may be a result of the significant time-gap between assessments (baseline, 3-months and 12-months), rather than a shortcoming in the instrument. Moreover, seasonal changes and other seasonal variation of work needs might have impacted the PSQI assessment over the three timepoints.

A systematic review by Manzar et al. (2018) showed that the factor structure of PSQI was best explained by two-factor models (13 out of 45 studies), followed by a one-factor model (9 out of 45 studies) and three-factor models (8 out of 45 studies) [9]. Three studies examining the factor structure of the PSQI have reported the same two-factor structure reported from our EFA as the most appropriate model-fit for the instrument across a range of samples including among breast cancer survivors in the US [12], the general population in Australia [37], and a multinational study involving students in Chile, Ethiopia and Thailand [42].

The PSQI is a widely used tool in clinical and research settings. However, more and more studies advocate a reconstruction of this unitary construct. A recent study proposed that PSQI is a better reflection of sleep on weekdays/workdays rather than weekends/free days, since the questions posed do not differentiate; responders are queried regarding their usual sleep, whereby the major time period is attributed to weekdays/workdays [43]. This is further evidence of the restrictive value of the global PSQI score alone. Therefore, studies focused on describing different understandings of the PSQI constructs and, perhaps contextual modifications are extremely relevant in order to increase the value of this tool in clinical practice and epidemiological studies. Since more and more studies advocate the multifactorial application of this instrument [9], it is pertinent to have evidence-based structures of this measurement. By considering distinct dimensional structures of sleep quality, future clinical studies are more likely to uncover individualized protocols for the management of sleep problems.

The present findings have important implications for the assessment of subjective sleep quality in working populations in Singapore. Although the PSQI is a well-validated measure of sleep quality, the current findings, in accordance with previous research, suggest that sleep quality may be better assessed with the multiple factors identified within the PSQI, as opposed to its single unidimensional measure. Such an approach may better identify workers experiencing sleep impairment that would otherwise go undetected with the unidimensional use of the PSQI and pure dichotomization of the scale into “good” and “poor” sleepers.

We have argued the case of the multidimensional nature of sleep quality and while the results of the present factor analysis suggest that a two- and three-factor model of the PSQI provides a better fit than the unidimensional measure, we acknowledge that the application and clinical utility of these models are unknown. For this reason, and given the clinical evidence and usefulness backing up the unidimensional application of the PSQI in its ability to screen for sleep impairment, caution is needed when recommending changes to the scoring of the PSQI. Future studies are needed to examine if the two-factor structure of the PSQI is present across additional working populations in Singapore and subsequent studies are then needed to identify the appropriate factor score cut-off points that will optimally detect sleep impairment, and whether the two-factor model is as or more useful as the unidimensional measure from a clinical perspective.

This study has several strengths and limitations. The strengths of this study are the following: we evaluated the factor structure of PSQI at multiple timepoints with the same population. The statistical analysis performed separately at the three timepoints provided the same finding that the two-factor structure is superior to the original single-factor model initially proposed. The majority of studies performing a factor analysis of the PSQI involve specific clinical populations, and the results from these studies may not be generalizable to a working population. However, our study is not without its limitations. Almost 80% of study participants were men, as the industries were comprised mainly of positions taken up by men such as engineers, technicians, and traffic controllers, which may limit the generalizability of the study findings to women; however, the EFA conducted with only female participants produced similar results and the same two factors as the EFA for the whole sample combined. In addition, while reflecting national representation, the majority of participants were Chinese and therefore the results of the analysis may be less generalizable to Malay and Indian participants.

## 5. Conclusions

In conclusion, this study demonstrated that the PSQI encompasses two factors (subjective sleep quality and sleep efficiency) in a working population in Singapore, using data over three timepoints. The CFA illustrated that a two-factor model provides an acceptable fit to the data across three timepoints and was highly superior to the single-factor model, and performed similarly to a three-factor model. Since the fit indices were similar between the two and three-factor models, and the two-factor model being the more parsimonious of the two, we recommend that the two-factor model be used for evaluating sleep characteristics among the working population in Singapore.

## Figures and Tables

**Figure 1 ijerph-16-04590-f001:**
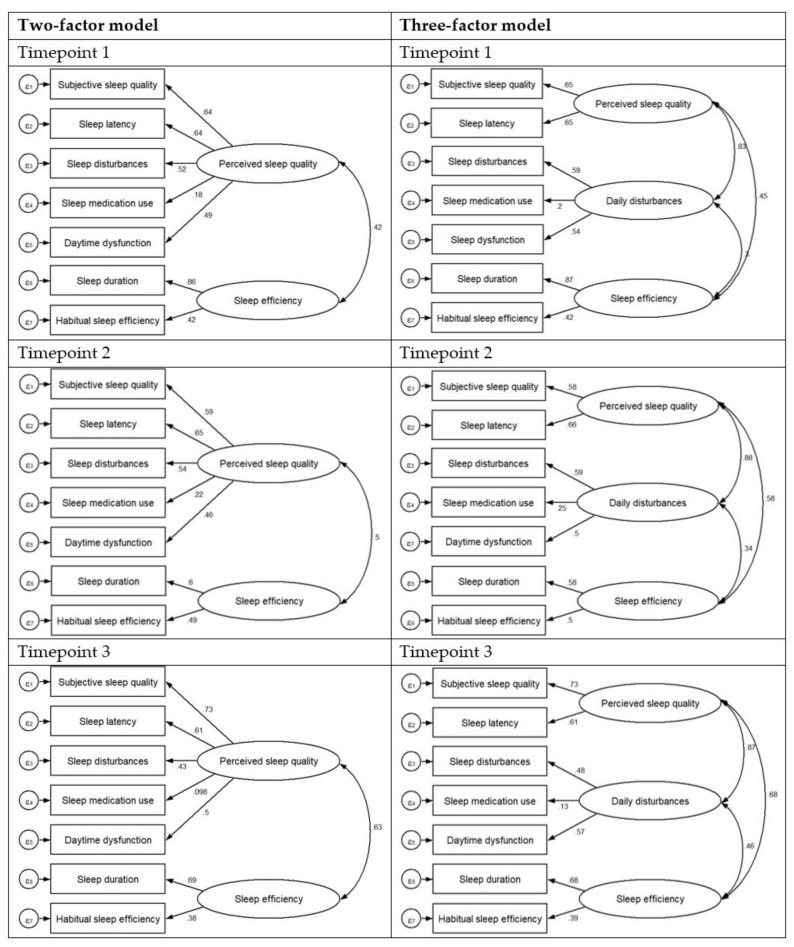
Factor loadings for the two and three-factor models at each timepoint.

**Table 1 ijerph-16-04590-t001:** Socio-demographic characteristics of the study cohort at each timepoint.

Variables	Timepoint 1 *n* = 464	Timepoint 2 *n* = 424	Timepoint 3 *n* = 329
**Age (years), (mean ± SD)**	39.0 ± 11.4	39.2 ± 11.3	40.7 ± 11.1
**Age (years), (*n*, %)**			
21–30	153 (33.0)	136 (32.1)	84 (25.5)
31–40	121 (26.1)	117 (27.6)	98 (29.8)
>40	190 (41.0)	171 (40.3)	147 (44.7)
**Gender, (*n*, %)**			
Male	369 (79.5)	334 (78.8)	256 (77.8)
Female	95 (20.5)	90 (21.2)	73 (22.2)
**Ethnicity, (*n*, %)**			
Chinese	296 (63.8)	271 (63.9)	216 (65.7)
Malays	99 (21.3)	89 (21.0)	60 (18.2)
Indians	48 (10.3)	44 (10.4)	39 (11.9)
Others ^a^	21 (4.5)	20 (4.7)	14 (4.26)
**Marital status, (*n*, %)**			
Single^b^	184 (39.7)	168 (39.6)	116 (35.3)
Married	280 (60.3)	256 (60.4)	213 (64.7)
**Education, (*n*, %)**			
Primary, secondary and higher secondary	116 (25.0)	103 (24.3)	86 (26.1)
Pre-college	183 (39.4)	172 (40.6)	123 (37.4)
College and above	165 (35.6)	149 (35.1)	120 (36.5)
**Monthly income, (*n*, %)**			
<S$4000	331 (71.3)	305 (71.9)	227 (69.0)
≥S$4000	133 (28.7)	119 (28.1)	102 (31.0)

Continuous variables are presented as mean ± standard deviation, and categorical variables as n (%); ^a^ Includes mixed ethnicities, Indonesians, Pakistanis and Filipinos; ^b^ Includes never married, widowed and divorced.

**Table 2 ijerph-16-04590-t002:** Pittsburgh Sleep Quality Index (PSQI) inter-component Spearman’s correlations and descriptive statistics for each timepoint.

		1	2	3	4	5	6	7	8
Timepoint 1	1. Subjective sleep quality	1							
	2. Sleep latency	0.42 **	1						
	3. Sleep duration	0.29 **	0.22 **	1					
	4. Habitual sleep efficiency	0.09 *	0.15 **	0.36 **	1				
	5. Sleep disturbances	0.27 **	0.36 **	0.10 *	0.10 *	1			
	6. Sleep medication use	0.09 *	0.07	0.08	0.06	0.08	1		
	7. Daytime dysfunction	0.32 **	0.25 **	0.20 **	0.04	0.31 **	0.14 **	1	
	8. Global PSQI	0.60 **	0.65 **	0.67 **	0.57 **	0.49 **	0.27 **	0.53 **	1
	Mean	1.04	0.94	1.03	0.61	1.10	0.07	0.69	5.48
	Standard deviation	0.55	0.85	0.93	0.98	0.51	0.37	0.66	2.76
	Median	1	1	1	0	1	0	1	5
	IQR	1–1	0–1	0–2	0–1	1–1	0–0	0–1	4–7
Timepoint 2	1. Subjective sleep quality	1							
	2. Sleep latency	0.39 **	1						
	3. Sleep duration	0.21 **	0.21 **	1					
	4. Habitual sleep efficiency	0.12 *	0.23 **	0.29 **	1				
	5. Sleep disturbances	0.26 **	0.40 **	0.09	0.15 **	1			
	6. Sleep medication use	0.14 **	0.09 *	0.01	−0.01	0.13 **	1		
	7. Daytime dysfunction	0.34 **	0.21 **	0.16 **	0.02	0.28 **	0.22 **	1	
	8. Global PSQI	0.58 **	0.67 **	0.60 **	0.60 **	0.54 **	0.29 **	0.53 **	1
	Mean	0.98	0.90	0.95	0.70	1.09	0.09	0.72	5.41
	Standard deviation	0.52	0.82	0.91	1.03	0.55	0.42	0.70	2.79
	Median	1	1	1	0	1	0	1	5
	IQR	1–1	0–1	0–2	0–1	1–1	0–0	0–1	3–7
Timepoint 3	1. Subjective sleep quality	1							
	2. Sleep latency	0.45 **	1						
	3. Sleep duration	0.32 **	0.29 **	1					
	4. Habitual sleep efficiency	0.19 **	0.18 **	0.26 **	1				
	5. Sleep disturbances	0.27 **	0.31 **	0.17 **	0.08	1			
	6. Sleep medication use	0.01	0.13 *	−0.01	0.01	0.11 *	1		
	7. Daytime dysfunction	0.41 **	0.24 **	0.19 **	0.06	0.26 **	0.09	1	
	8. Global PSQI	0.67 **	0.70 **	0.65 **	0.50 **	0.52 **	0.24 **	0.55 **	1
	Mean	1.01	0.86	0.97	0.42	1.08	0.09	0.64	5.08
	Standard deviation	0.58	0.87	0.94	0.80	0.55	0.42	0.67	2.77
	Median	1	1	1	0	1	0	1	5
	IQR	1–1	0–1	0–2	0–1	1–1	0–0	0–1	3–7

* *p* < 0.05; ** *p* < 0.01; IQR: inter-quartile range; PSQI; Pittsburgh Sleep Quality Index.

**Table 3 ijerph-16-04590-t003:** Factor Matrix for the two-factor solutions at each timepoint.

PSQI Subscales	Perceived Sleep Quality	Sleep Efficiency
Timepoint 1		
Subjective sleep quality	0.67 ^b^	0.23 ^f^
Sleep latency	0.69 ^b^	0.19 ^f^
Sleep duration	0.20 ^f^	0.79 ^a^
Habitual sleep efficiency	−0.02 ^f^	0.84 ^a^
Sleep disturbances	0.71 ^a^	−0.02 ^f^
Sleep medication use	0.29 ^f^	0.04 ^f^
Daytime dysfunction	0.67 ^b^	0.03 ^f^
Percentage of total variance, %	32.1	16.7
Timepoint 2		
Subjective sleep quality	0.64 ^b^	0.26 ^f^
Sleep latency	0.55 ^c^	0.45 ^d^
Sleep duration	0.14 ^f^	0.68 ^b^
Habitual sleep efficiency	0.01 ^f^	0.77 ^a^
Sleep disturbances	0.61 ^c^	0.22 ^f^
Sleep medication use	0.56 ^c^	−0.32 ^e^
Daytime dysfunction	0.71 ^a^	−0.04 ^f^
Percentage of total variance, %	31.6	17.2
Timepoint 3		
Subjective sleep quality	0.66 ^b^	0.37 ^e^
Sleep latency	0.68 ^b^	0.23 ^f^
Sleep duration	0.36 ^e^	0.63 ^b^
Habitual sleep efficiency	0.12 ^f^	0.68 ^b^
Sleep disturbances	0.66 ^b^	−0.07^f^
Sleep medication use	0.42 ^e^	0.52 ^d^
Daytime dysfunction	0.66 ^a^	0.02 ^f^
Percentage of total variance, %	32.8	15.7

Factor analysis conducted with maximum likelihood estimation extraction and direct oblimin rotation. ^a^ excellent loadings, ^b^ very good, ^c^ good loading, ^d^ fair loading, ^e^ poor loading, ^f^ loading too low to interpret.

**Table 4 ijerph-16-04590-t004:** Goodness-of-fit comparison of models.

Timepoint	Model	Chi-Square (*p*-Value)	GFI	AGFI	CFI	TLI	RMSEA	SRMR	CAIC	BIC
Timepoint 1	1-factor model ^a^	83.97 (<0.001)	0.79	0.68	0.81	0.72	0.10	0.06	109.63	6335.88
	2-factor model ^b^	36.61 (<0.001)	0.91	0.85	0.94	0.90	0.06	0.03	62.28	6294.66
	3-factor model ^c^	28.46 (0.003)	0.93	0.86	0.95	0.91	0.06	0.03	54.12	6298.79
Timepoint 2	1-factor model ^a^	67.22 (<0.001)	0.80	0.69	0.83	0.75	0.10	0.06	92.61	5969.28
	2-factor model ^b^	44.33 (<0.001)	0.87	0.79	0.90	0.84	0.08	0.04	69.72	5952.44
	3-factor model ^c^	37.37 (<0.001)	0.89	0.79	0.92	0.84	0.08	0.04	62.77	5957.59
Timepoint 3	1-factor model ^a^	31.23 (0.005)	0.89	0.83	0.93	0.90	0.06	0.04	55.85	4553.3
	2-factor model ^b^	21.31 (0.067)	0.92	0.88	0.97	0.95	0.04	0.03	45.93	4549.17
	3-factor model ^c^	17.27 (0.100)	0.94	0.88	0.98	0.95	0.04	0.03	41.89	4556.73

GFI goodness-of-fit index, AGFI adjusted goodness-of-fit index, CFI comparative fit index, TLI Tucker Lewis index, RMSEA root mean square error of approximation, SRMR Standardized Root Mean Square Residual, BIC Bayesian information criteria, CAIC consistent akaike information criteria, χ^2^ chi-square goodness-of-fit statistics; ^a^ The proposed single-factor model of Buysse et al; ^b^ Best model obtained from EFA for the study; ^c^ The proposed three-factor model of Koh et al.

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
