# Peer review of "Examining the Factor Structure of the Pittsburgh Sleep Quality Index in a Multi-Ethnic Working Population in Singapore"

_ijerph, 2019, doi:10.3390/ijerph16234590_

Round 1
Reviewer 1 Report
This study is aimed to investigate the dimensionality of the underlying factor structure of the Pittsburgh Sleep Quality Index (PSQI), administered on 3 occasions (baseline, 3 months and 12 months). It has been carried out in a multi-ethnic working population in Singapore. Exploratory factor analysis (EFA) found two latent factors at each timepoint, and the confirmatory factor analysis (CFA) compared this two-factor solution to a single factor and to a three-factor model. The results of the CFA suggest that the two and three-factor models were both superior to the unidimensional model.
It is an interesting psychometric study on the multidimensionality of the PSQI, and my opinion is -on the whole quite positive.
I only found a major limitation. The authors have three series of data points, i.e., baseline, 3 months and 12 months, but I did not find any evaluation of the test-retest stability of the PSQI. This analysis is mandatory and it should be carried out for the whole test and for the two factors showed by the EFA.
As a minor comment, I found that the sample was largely unbalanced with respect to the gender and ethnicity. This should be mentioned as a partial limitation and, above all, some information (i.e., control analyses) have to be performed to assess if the results of the EFA are different when considering these two factors.
Author Response
Reviewer Summary
This study is aimed to investigate the dimensionality of the underlying factor structure of the Pittsburgh Sleep Quality Index (PSQI), administered on 3 occasions (baseline, 3 months and 12 months). It has been carried out in a multi-ethnic working population in Singapore. Exploratory factor analysis (EFA) found two latent factors at each timepoint, and the confirmatory factor analysis (CFA) compared this two-factor solution to a single factor and to a three-factor model. The results of the CFA suggest that the two and three-factor models were both superior to the unidimensional model.
It is an interesting psychometric study on the multidimensionality of the PSQI, and my opinion is -on the whole quite positive.
Reply
Thank you for the positive feedback and allowing us the opportunity to submit a revised draft of our manuscript. We appreciate the time and effort that you have dedicated to providing your valuable feedback on our manuscript. We are very grateful for the comments you've provided on our paper. We have been able to incorporate changes to reflect all of the suggestions provided. We have highlighted the changes within the manuscript in yellow. Here is a point-by-point response to your comments and concerns.
Comment 1
I only found a major limitation. The authors have three series of data points, i.e., baseline, 3 months and 12 months, but I did not find any evaluation of the test-retest stability of the PSQI. This analysis is mandatory and it should be carried out for the whole test and for the two factors showed by the EFA.
Reply
Thank you for pointing this out. We have now run test-retest reliability of the global PSQI score and the factors from the EFA analysis. This addition has meant changes were added to the methods, results and discussion sections.
In the methods section (Line 110-111), we have added the following text:
“We assessed the test-retest reliability of the global PSQI score and factors from the EFA by intra-cluster coefficient (ICC).”
In the results section (Line 174-176), we have added the following text:
The ICC values for the global PSQI score across the three timepoints was .62, while it was .64 and .38 for the factors perceived sleep quality and sleep efficiency, respectively.
In the discussion section (Line 283-289), we have added the following text:
The test-retest reliability of the PSQI instrument and the factors ‘perceived sleep quality’ and ‘sleep efficiency’ was lower than the desired level (≥.80) for widely used instruments [39]. Given that adequate test-retest reliability of the PSQI has been reported in previous studies with shorter time-gaps between assessments (2-45 days)[40-42], and the potential for sleep quality to change, it is possible that the low test-retest reliability observed in this study may be a result of the significant time-gap between assessments (baseline, 3-months and 12-months) rather than a shortcoming in the instrument. Moreover, seasonal changes and other seasonal variation of work needs might have impacted the PSQI assessment over the three timepoints.
Comment 2
As a minor comment, I found that the sample was largely unbalanced with respect to the gender and ethnicity. This should be mentioned as a partial limitation and, above all, some information (i.e., control analyses) have to be performed to assess if the results of the EFA are different when considering these two factors.
Reply
Thank you for the comment. We have re-run the exploratory factor analysis with only female participants included in the analysis. The results from the EFA for the subgroup of female participants only were similar to the EFA for the whole sample. We have added the below text on Line 198-201.
“The results from the sensitivity analysis involving only female participants, produced similar results, and the same two factors as the EFA for the whole sample combined. The results for the factor matrix for the two-factor solutions at each timepoint for female participants only is provided in Table S2 of the supplementary files”
In addition, we have mentioned the gender imbalance in the limitations section on Lines 331-335, as below.
“Almost 80% of study participants were men, as the industries comprised mainly of positions taken up by men such as engineers, technicians, and traffic controllers which may limit the generalisability of the study findings to women, however, the EFA conducted with only female participants produced similar results and the same two factors as the EFA for the whole sample combined”
We have also acknowledged the imbalance regarding the ethnicity of participants as a limitation on Line 335-337 as below.
In addition, the majority of participants were Chinese and therefore the results of the analysis may be less generalisable to Malay and Indian participants.
However, we also acknowledge that the proportions of ethnicities in the study sample reflects the national representation of ethnicities in Singapore. To make this clearer, we have added the below text to the “Characteristics of Study Participants” section of the results (Line 152-154).
“Reflecting national representation, a large proportion were Chinese (63.8%), followed by Malays (21.3%), Indians (10.3%), and other Asian groups (4.5%).”
Reviewer 2 Report
This manuscript presents a factor analysis of the Pittsburgh Sleep Quality Index, a widely used scale that quantifies habitual sleep quality as a single dimension. The authors find that a two-factor (subjective sleep quality and habitual sleep efficiency) model of the PSQI was superior to the single dimensioned original PSQI version. A strength of the paper is the comparison against a three-factor model, and the use of exploratory and confirmatory factor analysis, both of which were highlighted in a recent review (Manzar et al 2018, Health Qual Life Outcomes) as being lacking from many investigations of the dimensionality of the PSQI. This previous review found 45 papers on the dimensionality of the PSQI, indicating that the topic of presently submitted manuscript is not especially novel, but it’s application in a specific working (non-clinical) cohort of interest is a worthwhile addition to the literature, particularly considering the methodological improvements over many earlier studies.
In principle I am happy to recommend this article for publication in IJERPH, although there were a small number of errors that should be corrected first:
Line 33: The acronym CFA is used, but not defined when mention on line 30
Line 113: Should define KMO acronym here, as it is used later on line 179
Line 165: According to Table 2, the lowest correlation (r=0.04) was between daytime dysfunction and habitual sleep efficiency, not sleep medication and habitual sleep efficiency (r=0.06) as stated in the text.
Line 178: Should be Table S2, not S1
Line 182, Table 2: It would be useful to indicate in the table caption that the values presented are correlation coefficients, so it can be interpreted without reading the text.
Author Response
Reviewer Summary
This manuscript presents a factor analysis of the Pittsburgh Sleep Quality Index, a widely used scale that quantifies habitual sleep quality as a single dimension. The authors find that a two-factor (subjective sleep quality and habitual sleep efficiency) model of the PSQI was superior to the single dimensioned original PSQI version. A strength of the paper is the comparison against a three-factor model, and the use of exploratory and confirmatory factor analysis, both of which were highlighted in a recent review (Manzar et al 2018, Health Qual Life Outcomes) as being lacking from many investigations of the dimensionality of the PSQI. This previous review found 45 papers on the dimensionality of the PSQI, indicating that the topic of presently submitted manuscript is not especially novel, but it’s application in a specific working (non-clinical) cohort of interest is a worthwhile addition to the literature, particularly considering the methodological improvements over many earlier studies.
In principle I am happy to recommend this article for publication in IJERPH, although there were a small number of errors that should be corrected first:
Reply
Thank you for the positive feedback and allowing us the opportunity to submit a revised draft of our manuscript. We appreciate the time and effort that you have dedicated to providing your valuable feedback on our manuscript. We are very grateful for the comments you've provided on our paper. We have been able to incorporate changes to reflect all of the suggestions provided. We have highlighted the changes within the manuscript in yellow. Here is a point-by-point response to your comments and concerns.
Comment 1
Line 33: The acronym CFA is used, but not defined when mention on line 30
Reply
Thank you for pointing this out. This has been amended on Line 30.
Comment 2
Line 113: Should define KMO acronym here, as it is used later on line 179
Reply
Thank you for pointing this out. This has been amended on Line 113.
Comment 3
Line 165: According to Table 2, the lowest correlation (r=0.04) was between daytime dysfunction and habitual sleep efficiency, not sleep medication and habitual sleep efficiency (r=0.06) as stated in the text.
Reply
Thank you for pointing this out. We now corrected our error, which is amended on Line 165.
Comment 4
Line 178: Should be Table S2, not S1
Reply
Thank you for pointing this out. This has been amended on Line 178.
Comment 5
Line 182, Table 2: It would be useful to indicate in the table caption that the values presented are correlation coefficients, so it can be interpreted without reading the text.
Reply
Thank you for point this out, the table was incorrectly labeled in the submitted version. It has now been amended and Table 2 is now labeled as “Pittsburgh Sleep Quality Index (PSQI) inter-component Spearman’s correlations and descriptive statistics for each timepoint”
Round 2
Reviewer 1 Report
The authors made the changes I requested and/or responded to the points I raised. In my opinion, it is now acceptable for publication